# Remix-DiT: Mixing Diffusion Transformers for Multi-Expert Denoising

**Gongfan Fang**   **Xinyin Ma**   **Xinchao Wang**[*]

National University of Singapore

{gongfan,maxinyin}@u.nus.edu, xinchao@nus.edu.sg

## Abstract

Transformer-based diffusion models have achieved significant advancements across a variety of generative tasks. However, producing high-quality outputs typically necessitates large transformer models, which result in substantial training and inference overhead. In this work, we investigate an alternative approach involving multiple experts for denoising, and introduce Remix-DiT, a novel method designed to enhance output quality at a low cost. The goal of Remix-DiT is to craft $N$ diffusion experts for different denoising timesteps, yet without the need for expensive training of $N$ independent models. To achieve this, Remix-DiT employs $K$ basis models (where $K < N$) and utilizes learnable mixing coefficients to adaptively craft expert models. This design offers two significant advantages: first, although the total model size is increased, the model produced by the mixing operation shares the same architecture as a plain model, making the overall model as efficient as a standard diffusion transformer. Second, the learnable mixing adaptively allocates model capacity across timesteps, thereby effectively improving generation quality. Experiments conducted on the ImageNet dataset demonstrate that Remix-DiT achieves promising results compared to standard diffusion transformers and other multiple-expert methods.

## 1   Introduction

Transformer-based Diffusion models [39, 8, 29, 2] have shown significant potential in generating high-quality images and videos [24, 20, 3]. However, achieving such quality often requires large transformer architectures [35], which incur considerable training and inference costs. To alleviate these computational burdens, multi-expert denoising has emerged as a promising approach [1, 18, 28, 16, 27], which employs multiple specialized diffusion models, each designed for distinct time intervals within the denoising process.

The goal of multi-expert denoising is to increase the overall capacity of diffusion models while keeping an acceptable overhead. The denoising process of diffusion models involves multiple different timesteps [13, 32, 18], requiring the network to make predictions at various noise levels. Previous work has shown that the tasks of diffusion models vary across different timesteps [13, 37, 27]. For instance, at higher noise levels, the model focuses more on low-frequency features, while at lower noise levels, the model emphasizes generating high-frequency details [37]. This variability inherently leads to a multi-task problem [16, 18]. However, due to the limited capacity of a single diffusion model, it is challenging to craft a comprehensive and balanced model that performs well across the entire denoising process. To alleviate this issue, multiple-expert denoising deploys specialized expert models at different timesteps. To this end, each model only needs to learn the denoising task for a specific subset of timesteps. Although this design introduces multiple models, only one model

---

[*]Corresponding Author

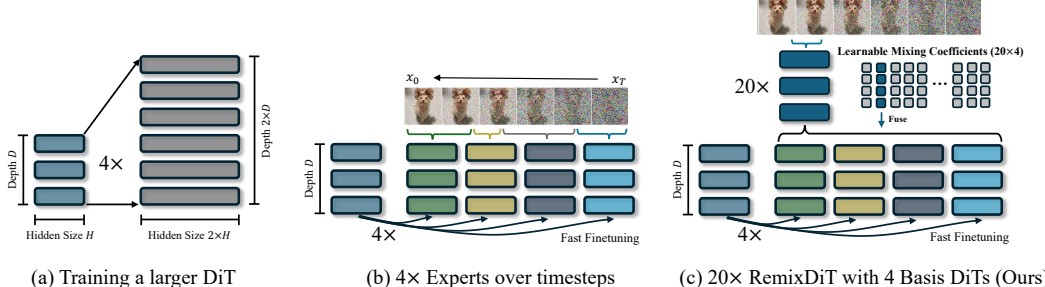

(a) Training a larger DiT  (b) 4× Experts over timesteps  (c) 20× RemixDiT with 4 Basis DiTs (Ours)

Figure 1: (a) Re-training a larger DiT incurs significant training and inference costs. (b) Multi-expert denoising trains multiple expert models to improve generation quality while maintaining low inference overhead. However, training multiple experts still results in significant training costs. (c) This work introduces RemixDiT, a learnable method to craft any number of experts by mixing basis models.

is activated at each timestep during the inference process, and therefore the overall computational complexity does not increase significantly.

In a multi-step denoising process, such as a 1000-step denoising chain, determining the optimal number of intervals and their partitioning remains an open problem. Existing approaches typically employ an even-split strategy, which distributes the denoising process equally among a predefined number of expert models. This uniform allocation often leads to suboptimal utilization of model capacity. In addition, the training of multiple expert models introduces substantial computational costs. Due to the inherent task similarity between adjacent intervals in the denoising process, training separate experts without accounting for this similarity can result in redundant and inefficient training. Although some recent methodologies, such as tree-structured training [1], have proposed hierarchical strategies to mitigate these issues, the process of training and capacity allocation remains largely manual and heuristic. To address these challenges, this paper introduces a novel, learnable strategy to craft multiple experts adaptively for denoising.

Consider a denoising process of $T$ steps. The objective of this paper is to construct $N$ expert models, each handling an interval of length $\frac{T}{N}$ as shown in Figure 1 (b). A straightforward approach to improve performance is to increase the number of experts, as more experts provide greater total capacity, leading to enhanced performance. However, this also results in a linear increase in training costs. The core idea of the proposed method lies in training only $K(K < N)$ basis models, which can be used to craft $N$ expert models efficiently for inference. To be exact, we construct $K$ transformer-based diffusion models and introduce learnable mixing coefficients to fuse their parameters, which can be easily implemented as an embedding layer. During training, the coefficients will learn to allocate the model capacity across different timesteps. During inference, we precompute those mixed models to do a chained multi-expert denoising.

We validated the effectiveness of our method on ImageNet-256x256 [6]. Owning to the learnable nature of Remix-DiT, our approach adaptively allocates model capacity across timesteps, thereby achieving superior generation results compared to independently trained multiple expert models. Our analysis of the learned mixing coefficients revealed several insightful findings: Firstly, the coefficients exhibit a natural similarity between adjacent timesteps, indicating that tasks at neighboring steps are relatively similar. Conversely, the coefficients for timesteps that are further apart show distinct differences, supporting the multi-expert denoising hypothesis that the learning target at different steps can be quite diverged. Additionally, our method's learnable coefficients tend to allocate more capacity to early timesteps, suggesting that the algorithm identifies predictions at lower noise levels as more challenging and thus requires greater capacity to generate finer details. In contrast, at higher noise levels (i.e., timesteps approaching $T$), the algorithm integrates multiple basis models for higher utilization of model capacity. Furthermore, we observed that experts created through hybridization exhibit specialization, achieving lower loss at specific timesteps while incurring higher loss at others. Through collaborative denoising, such specialization reduces overall prediction errors, highlighting the effectiveness of our method.

The contribution of this work lies in a novel method for multi-expert denoising, which creates N experts by combining K (K<N) basis models. This approach effectively reduces training costs and improves performance.

## 2 Related Works

**Multiple Experts in Diffusion Models** The multi-step denoising process inherent to the diffusion model [14, 33, 34] can be viewed as a multitasking problem [1, 18, 28, 16]. At each step, the model receives inputs with varying levels of noise and makes predictions accordingly. Given the disparate learning objectives for each denoising step, the utilization of a singular model across all stages is, to a certain extent, inefficient and challenging. Several prior works suggested employing multiple models, each dedicated to handling partial timestep intervals. For example, E-Diff [1] introduces a simple binary strategy to learn various experts and during inference, take the ensemble of experts for prediction. OMS-DPM [18] proposes a model scheduler, dynamically taking denoiser of different sizes from a model zoo for inference. MEME [16] deploys a similar paradigm, yet training lightweight models for different steps. The ensemble of these lightweight denoisers can achieve superior performance compared to their larger counterparts while maintaining efficiency during inference. Another exploration to enhance efficiency using multiple experts is T-Stich [27], which stitches the denoising trajectories of different pre-trained models, enabling dynamic inference cost without additional training. However, it's noteworthy that multiple diffusion models may incur additional storage and context switch costs. Thus, DTR [28] introduces a novel multi-tasking strategy to learn a single model with scheduled task masks, activating specific sub-networks for different timesteps.

**Transformer-based Diffusion Models** Transformer-based diffusion models have achieved impressive results on image generation [2, 29, 4, 10, 20, 17]. DiT [29] explores the scalability of transformers for image generation, achieving competitive performance compared to CNN-based diffusion models. UViT [2] independently explores transformer design for generation, incorporating the U-Net [31, 22, 30] architecture for improved training efficiency and quality. Furthermore, recent works have expanded transformer-based diffusion models to video generation [20, 21, 24, 5, 3], audio [19, 15] and 3D [25] which shows the power of transformers in modeling complicated data. However, training diffusion models remains inefficient, typically requiring millions of steps to produce satisfactory results. A series of works in the literature focus on the efficiency of diffusion-based transformers [40, 40, 23, 9]. MaskDiT [40] and MDTv2 [10] utilize masked transformers and an additional masked autoencoder task [12] to streamline training. UViT [2] also demonstrates the benefits of skip connections for accelerating convergence. In this work, we introduce a new strategy to enhance the training efficiency of plain diffusion transformers

## 3 Preliminary

**Diffusion Probabilistic Models** Diffusion Probabilistic Models (DPMs) train a denoiser network to revert a process of adding gradually increased noises [13]. Given an input $x_0$, a $T$-step *forward process* is deployed to transform $x_0$ into latent $x_1, \ldots x_T$, where the transformation is defined as $q(\boldsymbol{x}_t|\boldsymbol{x}_{t-1}) = \mathcal{N}(\boldsymbol{x}_t; \sqrt{1-\beta_t}\boldsymbol{x}_{t-1}, \beta_t I)$ under a increased variance schedule $\beta_{1:T}$. To revert this process, a network $q(\boldsymbol{x}_{t-1}|\boldsymbol{x}_t)$ is trained to predict $x_{t-1}$ given $x_t$. By defining $\alpha_t = 1 - \beta_t$ and $\bar{\alpha}_t = \prod_{s=1}^{t} \alpha_s$, the training objective is formalized as

$$\mathcal{L}(\boldsymbol{\theta}) = \mathbb{E}_{t,\boldsymbol{x}_0,\boldsymbol{\epsilon}\sim\mathcal{N}(0,1)} \left[ \|\boldsymbol{\epsilon} - \boldsymbol{\epsilon_\theta}(\sqrt{\bar{\alpha}_t}\boldsymbol{x}_0 + \sqrt{1-\bar{\alpha}_t}\boldsymbol{\epsilon}, t)\|^2 \right], \tag{1}$$

where $\epsilon_\theta$ is a trainable neural network and $\epsilon$ refer to the noises at timestep $t$. A common practice in the literature is to train a single neural network for all timesteps $t$. However, it has been observed that the denoising behavior usually varies across timesteps [13, 1, 37, 27]. Due to the limited capacity, a single model may struggle to fit all steps. Therefore, a more suitable yet natural approach is to use multiple models specifically learned for different timestep intervals [1]."

**Multi-Expert Denoising** To facilitate denoising with $N$ expert models, the timesteps are typically divided into $N$ intervals. Each model, denoted as $\epsilon_{\theta_i}$, contains independent parameters $\boldsymbol{\theta}_i$. For each model $\boldsymbol{\theta}_i$, we minimize the training objective as formalized in Equation 1 on the corresponding

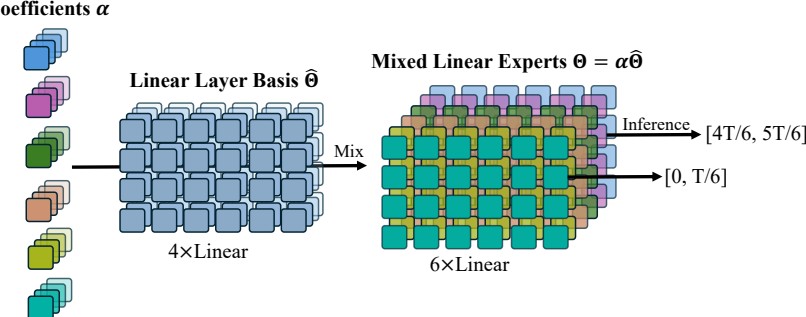

Figure 2: An example of mixing 4 linear layers basis into 6 expert layers. Each expert linear layer is a weighted averaging of the basis layers. At each denoising interval, only one expert is activated for inference or training. To increase the number of experts, we increase the number of coefficients $\boldsymbol{\alpha}$, which is more efficient than independently training new experts.

timestep interval $t \in [i \times \lfloor \frac{T}{N} \rfloor, (i+1) \times \lfloor \frac{T}{N} \rfloor]$ as follows:

$$\mathcal{L}(\boldsymbol{\theta}_i) := \mathbb{E}_{t \in [i \times \lfloor \frac{T}{N} \rfloor, (i+1) \times \lfloor \frac{T}{N} \rfloor], \boldsymbol{x}_0, \boldsymbol{\epsilon} \sim \mathcal{N}(0,1)} \left[ \| \boldsymbol{\epsilon} - \boldsymbol{\epsilon}_{\boldsymbol{\theta}_i}(\sqrt{\bar{\alpha}_t} \boldsymbol{x}_0 + \sqrt{1 - \bar{\alpha}_t} \boldsymbol{\epsilon}, t) \|^2 \right]. \tag{2}$$

Although the simple form of multi-expert denoising has been verified effective in previous works [1, 16], the above objective still presents two major challenges: 1) The optimal number of experts and the best interval partition is unknown; 2) Training multiple independent experts can be expensive. In this work, we introduce a new approach, Remix-DiT to address the above issues.

## 4   Method

**Uniform Partition of Denoising Process**    Let's consider a multi-expert denoising problem with $N$ experts for $T$ timesteps. For simplicity, we assume that each expert's parameters $\boldsymbol{\theta}_i$ is a column vector with $P$ trainable elements and the parameter matrix of all experts can be formalized as $\boldsymbol{\Theta}_{N \times P} = [\boldsymbol{\theta}_1, \boldsymbol{\theta}_2, \cdots, \boldsymbol{\theta}_N]^\top$. This is natural since the parameters in a model can be flattened and concatenated into a vector. We consider a simple partition of timesteps, which equally divides the denoising process into $N$ intervals, i.e., $[0, T/N] \dots [(N-1) \cdot T/N, N \cdot T/N]$. The motivation behind the oracle partition is that adjacent steps share similar noise levels and training objectives. Following Equation 2, this leads to the learning objective to optimize each expert on their associated time intervals.

$$\boldsymbol{\Theta}^* = \arg \min_{\boldsymbol{\Theta}} \sum_{\boldsymbol{\theta}_i \in \boldsymbol{\Theta}} \mathcal{L}(\boldsymbol{\theta}_i). \tag{3}$$

The above process involves $N$ independent optimization problem, which presents some issues. First, the optimal number of experts is unknown. It's difficult to increase the number of experts since this will introduce huge training costs. Besides, the optimal partition is also unclear, making the uniform partition inefficient. In this work, we investigate such a problem: Is it possible to learn any number of experts, while keeping an affordable overhead?

**Crafting Experts by Mixing**    To address the problem of uniform partition, we leverage a set of basis models to avoid straightforward training on experts. The core idea lies in that, it is possible to fuse the parameters of two diffusion models to achieve better performance [38]. Formally, instead of training $N$ independent models directly, the core idea of Remix-DiT is to learn $K$ $(K < N)$ basis models with the same architecture as experts, parameterized by $\boldsymbol{\beta}_{K \times P} = [\boldsymbol{\beta}_1, \boldsymbol{\beta}_2, \dots \boldsymbol{\beta}_K]^\top$. And the expert models can be crafted by mixing the basis parameters with certain coefficients. For each expert $\boldsymbol{\theta}_i$, we associate it with a coefficients $\boldsymbol{\alpha}_i = [\alpha_{i1}, \alpha_{i2}, \dots \alpha_{iK}]^\top$ and compute the mixed expert parameter through a weighted averaging $\boldsymbol{\theta}_i = \sum_k \alpha_{ik} \boldsymbol{\beta}_k$. This leads to the coefficient matrix denoted as $\boldsymbol{\alpha}_{N \times K} = [\boldsymbol{\alpha}_1, \boldsymbol{\alpha}_2, \dots, \boldsymbol{\alpha}_N]^\top$. Then, the expert's parameter matrix can be easily obtained through a simple matrix multiplication:

$$\boldsymbol{\Theta}_{N \times P} = \boldsymbol{\alpha}_{N \times K} \boldsymbol{\beta}_{K \times P}. \tag{4}$$

With Equation 4, we can freely craft different experts using different mixing coefficients. Now the problem lies in that how to learn a good coefficients $\alpha$ and basis models $\beta$.

**Architecture of Remix-DiT** Let's further delve into the details of $K$ basis models, denoted as $\beta$, alongside a coefficients matrix $\alpha$. In this study, we adopt the DiT architecture proposed in [29], which primarily consists of linear and normalization layers. Rather than constructing $K$ distinct models, we propose a simple trick to construct a singular and extended DiT model to encapsulate all $K$ basis models [36]. This is achieved by increasing the width of the linear layers by a factor of $K$, with a modified forward process to support the mixing of basis. An example of the extended linear

---

**Algorithm 1** Mixed Linear (PyTorch-like Pseudo Code)

```
1: function MIXEDLINEAR(x, w, coeffs, K)
2:     // x: linear inputs
3:     // w: linear parameters
4:     // coeffs: mixing coefficients
5:     // K: number of basis
6:     w = w.view(K, w.shape[1]//K, -1)
7:     w_ = (coeffs * w).sum(0)
8:     return torch.nn.functional.linear(x, w_)
9: end function
```

---

layer is depicted in Algorithm 1. A notable distinction between the standard DiT and the proposed Remix-DiT is the integration of a weighted averaging step before the true computation. Despite the expansion in width by a factor of $K$, the effective weights engaged during forward propagation are equivalent to those of a normal DiT of the original width. The additional computational cost is attributed solely to the element-wise operation during mixing. Furthermore, during the backward propagation phase, the computational cost of the mixed network almost mirrors those of a standard DiT, except for the additional overhead caused by the mixing operation.

**Network Training** A notable challenge presented by Remix-DiT is the restriction of activating only one mixed expert during each forwarding and backwarding phase. To address this limitation, we employ a hierarchical sampling strategy whereby the expert $\theta_i$ is selected first, followed by a random selection of timestep $t$ within the interval $[i \times \lfloor \frac{T}{N} \rfloor, (i+1) \times \lfloor \frac{T}{N} \rfloor]$. Consequently, only one expert is activated per training step. Despite this constraint, a critical advantage of Remix-DiT is that regardless of which expert is activated, all basis models remain updatable. This feature distinctly sets our approach apart from methodologies that train experts independently. The training and inference processes of Remix-DiT are delineated in Algorithm 2. The training protocol for Remix-DiT closely mirrors that of a standard DiT, with the exception that the sampled timesteps $t$ must originate from a singular time interval during each step. During inference, the compact nature of the basis models allows for reduced GPU memory usage compared to maintaining $N$ separate experts, providing a significant advantage. Additionally, it is still feasible to pre-compute the $N$ experts for more efficient inference without runtime mixing.

**Transform Pre-trained DiT to Remix-DiT with Prior Coefficients** Training generative models from scratch is usually inefficient. Note that the proposed method Builds a DiT model with $K$-times width. This allows it to inherit the weights of pre-trained DiTs, by replicating the pre-trained weight $K$ times. This leads to $K$ basis models with the same parameters. However, this will be problematic if we inspect the gradient to each mixing coefficient, denoted as $\nabla_{\alpha_{i,k}}$:

$$\nabla_{\alpha_{i,k}} = \sum_k \nabla_{\theta_i} \odot \beta_k, \tag{5}$$

where $\odot$ is an element-wise product of two matrices. First, the update of basis parameter $\hat{\theta}_i$ is usually slow, this makes all mixed experts similar to each other at the beginning of training and also makes the gradient w.r.t the mixed expert $\hat{\theta}_i$ almost unchanged. In this case, the gradient to the coefficients will be constant. To address this issue, we introduce a prior regularization to force each basis $\hat{\theta}_i$ to learn different objectives. Inspired by the oracle partition in multiple expert denoising, we craft prior coefficients $\alpha_*$ with one-hot vectors. For example, the hand-coded prior coefficient $[0, 1, 0, 0]$ directly assigns the second basis as the expert, this encourages a polarization for different bases. And the regularization can be easily implemented as a cross-entropy between the prior coefficients and the learnable coefficients.

$$\mathcal{R} = -\gamma \sum_i \alpha_{i,k}^* \log(\alpha_{i,k}), \tag{6}$$

---

**Algorithm 2** Remix-DiT

 1: // Training
 2: Initialize $K$ basis models $\boldsymbol{\beta} = [\boldsymbol{\beta}_1, \boldsymbol{\beta}_2, \ldots \boldsymbol{\beta}_K]^\top$ from random or pretrained models
 3: Randomly initialize mixing logits $\boldsymbol{\pi} = [\boldsymbol{\pi}_1, \boldsymbol{\pi}_2, \ldots \boldsymbol{\pi}_N]^\top$
 4: **while** Training not terminated **do**
 5:     Randomly sample a expert index $i \in [0, N)$
 6:     Compute the mixing coefficients $\boldsymbol{\alpha}_i = \text{Softmax}(\boldsymbol{\pi}_i)$
 7:     Obtain the $i$-th mixed expert $\boldsymbol{\theta}_i = \boldsymbol{\alpha}_i \boldsymbol{\beta}$
 8:     Randomly sample the timestep $t \in [i \times \lfloor \frac{T}{N} \rfloor, (i+1) \times \lfloor \frac{T}{N} \rfloor]$
 9:     Compute the loss $\mathcal{L}_{t,i} = \|\boldsymbol{\epsilon} - \boldsymbol{\epsilon}_{\boldsymbol{\theta}_i}(\sqrt{\bar{\alpha}_t}\boldsymbol{x}_0 + \sqrt{1 - \bar{\alpha}_t}\boldsymbol{\epsilon}, t)\|^2$
10:     Update $\boldsymbol{\pi}$ and $\boldsymbol{\Theta}$ by back-propagation
11: **end while**
12: // Inference
13: Compute the mixing coefficients $\boldsymbol{\alpha} = [\text{Softmax}(\boldsymbol{\pi}_1), \text{Softmax}(\boldsymbol{\pi}_2), \ldots, \text{Softmax}(\boldsymbol{\pi}_N)]$
14: Pre-compute experts with $\boldsymbol{\Theta} = \boldsymbol{\alpha}^T \boldsymbol{\beta}$
15: Inference with $\boldsymbol{\Theta}$

---

where $\gamma$ is a hyperparameter to control the strength of regularization. In practice, we adopt an annealing strategy to linearly remove the regularization with $\gamma \to 0$.

## 5 Experiments

### 5.1 Experimental Settings

**Network Architecture.** This paper utilizes the DiT [29] models as the fundamental architecture. By reloading the computation process of the linear layers, additional mixing coefficients is introduced to create RemixDiT. For a fair comparison, we ensure that the dimensions of all mixed experts are completely consistent with the original DiT. For instance, Remix-DiT-S is constructed by quickly combining multiple DiT-S models to form an expert model.

**Training Details.** Since our mixed model is entirely consistent with the standard DiT, this method can be applied to pre-trained models. We first trained a standard DiT model on ImageNet and then initialized the basis model with the same pre-trained weights. We introduce prior as discussed in Equation 6 to the mixing coefficients to accelerate the learning of different basis models. In our experiments, we conducted 100 K fine-tuning on DiT-S/B/L models [29], pre-trained for 2M/1M/1M steps correspondingly.

### 5.2 Transform Pretrained DiT to Remix-DiT

In Table 1, we present the results of fine-tuning standard DiT models using the Remix-DiT approach. Given the architectural consistency between the mixed model and the standard DiT, it is feasible to initialize $K$ basis models within the standard DiT framework. This enables the construction of RemixDiT with minimal additional training steps since the pre-trained DiT has been comprehensively trained at all temporal steps. For instance, starting with a pre-trained DiT-B model, our method successfully crafts a Remix-B with only 100K training steps, achieving superior performance compared to both the original models and baselines such as continual training and multiple experts [1].

Moreover, within the same computational budget, our approach outperforms Multi-expert baselines, where experts are trained independently. It is noteworthy that for Multi-expert baselines with eight experts, the total training budget is uniformly allocated to each expert. As the number of experts increases, the allocated training steps for each expert become more limited. In contrast, Remix-DiT trains shared basis models throughout the entire training process, allowing each model to be sufficiently updated.

As discussed earlier, RemixDiT allocates gradients from different time steps by employing mixing coefficients. Therefore, for $K$ basis models initialized similarly, greater diversity in the initial mixing coefficients facilitates the rapid learning of different models. To achieve this, we sequentially distribute the entire denoising process among the basis models. During subsequent fine-tuning, the

| Conditional Diffusion Transformers (DiT) - ImageNet 256×256 - cfg=1.5 | | | | | | |
|---|---|---|---|---|---|---|
| Method | #Effective Params | #Gflops per step | IS | FID | Precision | Recall |
| DiT-L/2-1M | 458.10 M | 80.79 | 196.34 | 3.73 | 0.8187 | 0.5398 |
| + 100k Cont. Training | 458.10 M | 80.79 | 200.16 | 3.57 | **0.8193** | 0.5356 |
| + 100k Multi Experts 8×L [1] | 8×458.10 M | 80.79 | 205.39 | 3.41 | 0.8149 | 0.5448 |
| **+ 100k Remix-L-4-20** | 4×458.10 M | 80.79 | **207.54** | **3.22** | 0.8175 | **0.5451** |
| DiT-B/2-1M | 130.51 M | 23.04 | 119.74 | 10.11 | 0.7348 | 0.5487 |
| + 100k Cont. Training | 130.51 M | 23.04 | 125.32 | 9.31 | 0.7418 | 0.5499 |
| + 100k Multi Experts 8×B [1] | 8×130.51 M | 23.04 | 126.58 | 9.28 | 0.7396 | 0.5504 |
| **+ 100k Remix-B-4-20** | 4×130.51 M | 23.04 | **127.42** | **9.02** | **0.7453** | **0.5553** |
| DiT-S/2-1M | 32.96 M | 6.07 | 48.46 | 32.55 | 0.5476 | 0.5677 |
| DiT-S/2-2M | 32.96 M | 6.07 | 59.39 | 26.51 | 0.5896 | 0.5613 |
| + 100k Cont. Training | 32.96 M | 6.07 | 60.34 | 26.05 | 0.5941 | 0.5672 |
| + 100k Multi Experts 8×S [1] | 8×32.96 M | 6.07 | 63.40 | 25.00 | **0.6017** | 0.5622 |
| **+ 100k Remix-S-4-20** | 4×32.96 M | 6.07 | **69.23** | **22.84** | 0.5998 | **0.5705** |

Table 1: Finetuning pre-trained DiT for 100k steps on ImageNet-256×256. During inference, all expert share the same architecture as a standard DiT.

variation in coefficients will enable a more refined allocation of model capacity. In Table 3, we compare the fine-tuned Remix-B to other diffusion models.

## 5.3 Visualization of Learned Experts

To further investigate the behavior of the experts learned by the algorithm, we visualized the mixing coefficients. As shown in Figure 4a, we conducted experiments on DiT-S. During the training of 20 expert models, we observed that the algorithm assigned more one-hot coefficients to timesteps close to 0. At these steps, the denoising model focuses more on high-frequency detail features. In contrast, at late timesteps resembling random noise, the algorithm enabled and mixed multiple models to generate content rather than using only one expert. As will be

| Method | #Params per Step | FID |
|---|---|---|
| ADM [7] | 554M | 10.94 |
| IDDPM [26] | 270M | 12.26 |
| VQ-Diffusion [11] | - | 11.89 |
| DiT-B/2 1M [29] | 130 M | 10.11 |
| MEME [16] | 103 M | 13.19 |
| Remix-B/2-4-20 | 130 M | 9.02 |

Figure 3: Comparision to existing methods

illustrated in Figure 5, mixing multiple models can improve the shape quality. Moreover, it can be observed that adjacent timesteps exhibit similar mixing coefficients, whereas more distant timesteps show greater differences. This finding supports the core hypothesis of multi-expert reasoning. To validate this observation, we further trained a RemixDiT with 8 experts. Figure 4b also shows the distribution of mixing coefficients, revealing results similar to the previous experiment. Additionally, Figure 4c visualizes the training loss functions of the 8 expert models throughout the denoising process. Each mixed expert has a lower loss function within its respective timestep intervals. The farther from an expert model's current interval, the higher its loss. Notably, at step 0, we found that the ensemble-constructed model effectively reduces prediction loss.

## 5.4 Analytical Experiments

In this section, we validate some key design aspects of RemixDiT. This includes the model's mixing method, the number of expert models and basis models, and whether the coefficients are independent or shared across layers.

**Model Mixers.** The core of the proposed method lies in the mixing of multiple models, and we explored three mixing methods: 1) Oracle Mixing: Basis models are manually assigned to different intervals as expert models, equivalent to training independent expert models, which makes the algorithm revert to the standard multi-expert training strategy regardless of N. 2) Raw Mixer: Each expert is assigned a real-valued coefficient without any constraints. 3) Softmax Mixer: This builds on the raw mixer by applying a softmax operation, ensuring that the mixed model is always a weighted average of the basis models. Table 2 shows the effectiveness of each mixer. We found that while oracle mixing improves model performance, it often fails to achieve optimal results due to manually

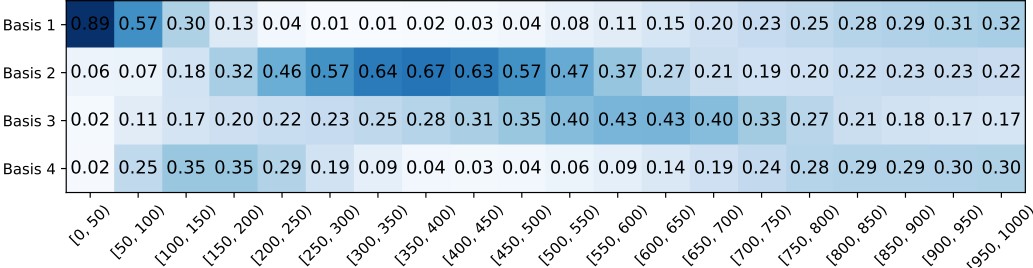

(a) Learned mixing coefficients for RemixDiT-S-4-20, which crafts 20 experts by mixing 4 basis models.

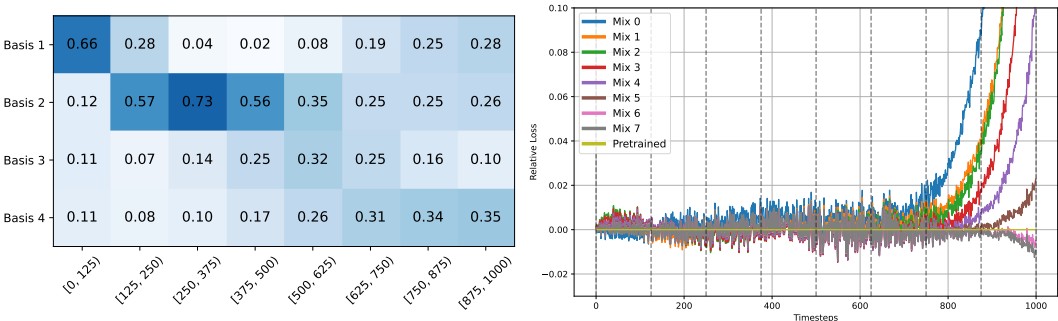

(b) Learned mixing coefficients for Remix-DiT-S-4-8

(c) Training losses of 8 experts over timesteps

Figure 4: (a) Learned Coefficients for a Remix-DiT-S-4-8, which mixes 4 basis DiT-S to obtain 8 models, each associated with a 125-step interval. The x-axis shows the corresponding timestep intervals associated with each mixed model. The value in each grid refers to the coefficients that will be used to weight the corresponding basis models. At early timesteps $T \to 999$, Remix-DiT tend to use an ensembled model for inference, which is averaged over all basis models. And at the late timesteps $T \to 0$, more specialized models, such as the first basis model are picked for fine-grained prediction.

designed partitions. The raw mixer, with its learnable coefficients, achieved better performance but included some impractical solutions, such as negative coefficients, which increased learning difficulty. The softmax mixer yielded the best results.

**Global or Local Mixer.** A DiT model typically comprises multiple layers. When mixing models, we can use globally shared mixing coefficients, referred to as the global mixer, or employ a layer-wise independent mixing strategy, referred to as the local mixer. The first strategy is simpler to train, as it requires only a small $K \times N$ coefficient matrix regardless of the network's depth. The second strategy, however, offers a larger model space by introducing trainable parameters that grow linearly with the number of layers. Despite this, the additional parameter size for both methods is very small compared to the original DiT model, which usually has tens to hundreds of millions of parameters. We validated both strategies in Table 2. However, we found that a simple global mixer yielded slightly better performance compared to the local mixer. For simplicity, we use a global mixer in our experiments.

**The Number of Experts and Basis** In this part, we also explored the impact of different numbers of basis and expert models on the algorithm's performance. It is evident that indefinitely increasing the number of expert models does not continuously enhance performance. There are three reasons for this: 1) First, the total capacity of the basis models is limited. With 4 basis models, it is difficult to craft 1,000 varied experts for denoising; 2) As illustrated in Figure 4a, some denoising steps share similar mixing coefficients, which limits the maximal number of effective experts in our method; 3) Due to the fact that only one expert is activated in forward and backward passes, the gradient w.r.t the mixing coefficients will be sparse. With a large coefficient table, the sparsity level of gradients will be higher, which introduces difficulty in optimization. However, compared to training the same $K$ expert models independently, our algorithm's advantage lies in its ability to learnably allocate the model's capacity through mixing coefficients. This allocation is done through a soft partition based on

| Ablation | #Params (M) | #Gflops per Step | IS-10K | FID-10K | Precision-10K | Recall-10K |
|---|---|---|---|---|---|---|
| **Mixer Type** | | | | | | |
| DiT-S/2 2M | 32.96 | 6.07 | 53.99 | 32.54 | 0.5691 | 0.6150 |
| - Onehot Mixer | 4×32.96 | 6.07 | 56.56 | 32.00 | 0.5726 | 0.5967 |
| - Raw Mixer | 4×32.96 | 6.07 | 57.39 | 31.46 | 0.5688 | **0.6122** |
| - Softmax Mixer | 4×32.96 | 6.07 | **57.93** | **31.11** | **0.5702** | 0.6071 |
| **Global or Local Mixer** | | | | | | |
| DiT-S/2 2M | 4×32.96 | 6.07 | 53.99 | 32.54 | 0.5691 | 0.6150 |
| - Global Mixer | 4×32.96 | 6.07 | **57.93** | **31.11** | 0.5702 | **0.6071** |
| - Local Mixer | 4×32.96 | 6.07 | 57.65 | 31.21 | **0.5712** | 0.6051 |
| **The Number of Basis and Expert Models** | | | | | | |
| DiT-S/2 2M | 4×32.96 | 6.07 | 53.99 | 32.54 | 0.5691 | 0.6150 |
| - Remix-S-4-4 | 4×32.96 | 6.07 | 56.47 | 31.77 | 0.5666 | 0.6059 |
| - Remix-S-4-8 | 4×32.96 | 6.07 | 57.15 | 31.60 | 0.5743 | 0.6109 |
| - Remix-S-4-20 | 4×32.96 | 6.07 | **57.93** | **31.11** | 0.5702 | 0.6071 |
| - Remix-S-4-100 | 4×32.96 | 6.07 | 57.00 | 31.48 | **0.5795** | **0.6111** |
| - Remix-S-4-1000 | 4×32.96 | 6.07 | 49.66 | 35.60 | 0.5571 | 0.6039 |
| - Remix-S-2-20 | 2×32.96 | 6.07 | 56.78 | 31.67 | 0.5703 | 0.6080 |
| - Remix-S-8-20 | 8×32.96 | 6.07 | 57.12 | 32.05 | 0.5642 | 0.6011 |

Table 2: Ablation with 10K fintuning over a pre-trained DiT-S/2. For efficiency, we compute the FID-10K with 100 sampling steps.

| Model | Steps per Sec | GPU Mem. (MiB) | Latency (Mixing) | Latency (pre-computed) |
|---|---|---|---|---|
| DiT-B/2 [29] | 2.93 | 13,152 | 15.77 ms | 15.77 ms |
| Remix-B/2-4-20 | 2.23 | 16,942 | 18.65 ms | 15.78 ms |
| Remix-B/2-4-8 | 2.29 | 16,926 | 18.54 ms | 15.82 ms |
| DiT-S/2 [29] | 3.40 | 9,758 | 5.66 ms | 5.66 ms |
| Remix-S/2-4-20 [29] | 3.24 | 10,578 | 7.16 ms | 5.71 ms |
| Remix-S/2-4-8 [29] | 3.30 | 10,558 | 7.04 ms | 5.69 ms |

Table 3: Training and inference efficiency of Remix-DiT. For inference, we can craft expert models by runtime mixing or pre-computing all experts, which is usually more efficient.

weighted loss, rather than a hard partition on individual models. This allows us to roughly choose an appropriate $N$ which is slightly larger than the optimal $N^*$, and the learnable coefficients can "merge" some experts if necessary by generating the same coefficients. For example, as shown in Figure 4a, we can find highly similar coefficients between the timestep intervals $[900, 950]$ and $[950, 1000]$, which means we don't need to allocate different experts in these timesteps. In our experiments, we craft 20 experts with 4 basis models.

**Inference and training Efficiency.** In Table 3, we present the training and inference efficiency of the proposed Remix-DiT. We found that the training speed slightly decreased from 2.93 to 2.29 due to the mixing of four basis models. Additionally, our method requires extra GPU memory to store the basis models. During inference, we estimated the latency of compact models that craft experts at runtime and a standard multi-expert denoising approach, where all experts are precomputed. Post-training, our method can achieve efficiency comparable to a standard DiT.

**Visualization of Generated Images.** Table 5 compares the generated images of the proposed RemixDiT-B and DiT-B. It can be observed that the shape of objects can be improved using the proposed methods. This is as expected since we allocate more model capacity to the early and intermediate stages of denoising, as illustrated in Figure 4, which mainly contributes to the image contents rather than details.

# 6 Limitations and Broader Impact

In this work, we introduce a learnable coefficient, implemented as a simple embedding layer for mixing. However, due to the limitations of the deep learning framework, we can only create one expert per forward and backward pass. This leads to sparse gradients in the embedding layers, where coefficients without gradients can only be updated with momentum rather than accurate gradients. One solution to alleviate this issue is distributed training, where processes craft different expert

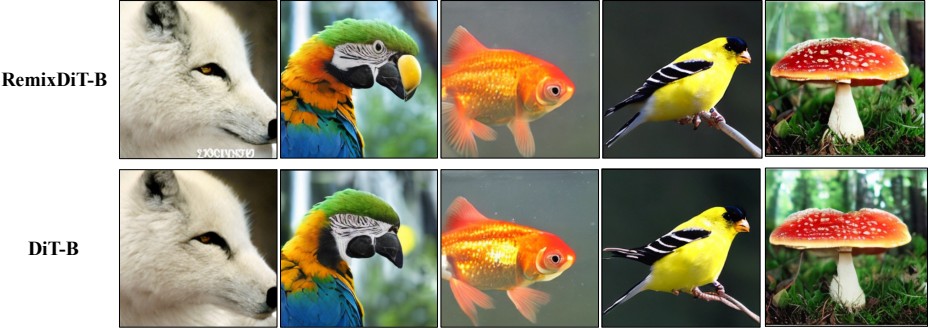

Figure 5: Visualization of generated samples from DiT-B and Remix-DiT-B.

models. Despite this, the challenge remains significant when training a Remix-DiT with a large number of experts, such as 1000. However, according to the visualization of learned coefficients, we find that 1000 experts may not be necessary since many adjacent timesteps share a similar model. In addition, this work will not introduce negative societal impact.

# 7 Conclusion

In this work, we introduce a multi-expert method to enhance the quality of transformer-based diffusion models while maintaining an acceptable inference overhead. The core contribution lies in the ability to craft a large number of experts from a few basis models, thereby significantly reducing the training effort. Besides, with our method, we don't have to accurately estimate the optimal number of required experts, since the learnable coefficients will adaptive merge experts if necessary, which brings huge flexibility to the practice.

# 8 Acknowledgment

This project is supported by the Ministry of Education, Singapore, under its Academic Research Fund Tier 2 (Award Number: MOE-T2EP20122-0006).

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
