# OpenReview forum: "Remix-DiT: Mixing Diffusion Transformers for Multi-Expert Denoising"
_NeurIPS.cc/2024/Conference — NeurIPS 2024 poster_

### Official Review · Reviewer_KY1f · 2024-07-04

**Soundness:** 3
**Presentation:** 3
**Contribution:** 2
**Rating:** 6
**Confidence:** 5

**Summary:**

This paper proposes Remix-DiT, which creates multiple experts by mixing fewer basis diffusion transformers, allowing each expert to specialize in the denoising task for corresponding timestep intervals. It achieves performance improvements by having each expert responsible for a larger number of timestep intervals with fewer total trainable parameters than previous multi-expert methods. Also, the paper analyzes the coefficients of how much each expert uses bases, demonstrating the denoising task similarity for adjacent timesteps, as well as the use of specialized bases for lower timesteps.

**Strengths:**

* The paper is structured well, making it easy to understand and follow.

* The proposed mixing basis strategy is interesting as it achieves better performance with fewer parameters compared to existing multi-expert methods.

* Ablation studies on mixing methods are comprehensive.

**Weaknesses:**

* **Lack of experiments.** The authors have to validate the performance of Remix-DiT by reporting comparisons with previous methodologies on the FFHQ or MS-COCO datasets. It would make the manuscript more solid if Remix-DiT achieves consistent performance improvements on multiple datasets.

* **Lack of comparison.** There are two methods, DTR [1] and Switch-DiT [2], to address the multi-task learning aspect of diffusion training by designing distinct denoising paths for 1000 timesteps in a single model. These are more parameter-efficient methods where they use no additional parameters or 10%, respectively. The authors should analyze them with respect to Remix-DiT.

[1] Park et al., Denoising Task Routing for Diffusion Models, ICLR 2024.

[2] Park et al., Switch Diffusion Transformer: Synergizing Denoising Tasks with Sparse Mixture-of-Experts, ECCV 2024.

**Questions:**

*  Is the Exponential Moving Average (EMA) model used to further train a pre-trained diffusion model?

* It would be better that the authors provide an affinity matrix between 20 timestep clusters based on the learned mixing coefficients. I think the affinity matrix could explain the similarity between denoising tasks.

---

> ### Author Rebuttal · Authors · 2024-08-07
>
> > **Q1: Lack of experiments. The authors have to validate the performance of Remix-DiT by reporting comparisons with previous methodologies on the FFHQ or MS-COCO datasets. It would make the manuscript more solid if Remix-DiT achieves consistent performance improvements on multiple datasets.**
>
> Thank you for the invaluable advice. We agree with the reviewer that incorporating more datasets is important to validate the robustness of RemixDiT and will add more datasets to polish this work. However, due to the limited rebuttal period, we are not able to present the results on large-scale datasets like COCO. We are currently working on FFHQ following DTR and should be able to update the results in the coming days once they are available.
>
> > **Q2: Lack of comparison. There are two methods, DTR [1] and Switch-DiT [2], to address the multi-task learning aspect of diffusion training by designing distinct denoising paths for 1000 timesteps in a single model. These are more parameter-efficient methods where they use no additional parameters or 10\%, respectively. The authors should analyze them with respect to Remix-DiT.**
>
> Thank you for the highly relevant pointers. We have supplemented the following results to compare RemixDiT to the mentioned baselines. We trained the proposed RemixDiT from scratch for 400K steps on ImageNet, which yielded a comparable FID to DTR. However, we also observed that the FID is higher than that of Switch-DiT-S, as training RemixDiT from scratch is less efficient, as discussed in Line 188 of our paper, where we have four basis models to build. Another reason for this might be the non-optimal hyperparameters for the scratch training. These baseline methods are very insightful and helpful. We will continue to improve this work to make our method more robust to different training configurations and will incorporate these results into the manuscript.
>
> | **Model**          | **Train Steps** | **FID** |
> |--------------------|-----------------|----------------|
> | DiT-S              | 400K            | 43.88          |
> | DTR-S [1]          | 400K            | 37.43          |
> | Switch-DiT-S [2]   | 400K            | 33.99          |
> | RemixDiT-S         | 400K            | 36.68          |
>
> [1] Park et al., Denoising Task Routing for Diffusion Models, ICLR 2024.
> [2] Park et al., Switch Diffusion Transformer: Synergizing Denoising Tasks with Sparse Mixture-of-Experts, ECCV 2024.

---

> > ### Comment · Reviewer_KY1f · 2024-08-08
> >
> > Thank you very much for your response. It well addressed my concerns. I will increase my rating to weak accept.

---

### Official Review · Reviewer_7vev · 2024-07-05

**Soundness:** 3
**Presentation:** 3
**Contribution:** 3
**Rating:** 7
**Confidence:** 4

**Summary:**

The paper introduces Remix-DiT, a modification to the diffusion transformer architecture that incorporates the multi-expert denoiser framework during both training and inference. Unlike traditional multi-expert methods that train $N$ separate individual experts independently for each time interval, Remix-DiT employs $K$ base models combined with $N$ mixing coefficients to dynamically compute time-specific experts. This approach enhances efficiency and leverages task similarities between adjacent intervals more effectively. Experiments on ImageNet demonstrate that Remix-DiT improves the performance of DiT across various model sizes.

**Strengths:**

- The paper is well-motivated and represents a valuable step towards integrating the multi-expert denoising framework into standard diffusion models.

- The main idea of the paper (using global mixers to compute the final experts) is novel and interesting to me in this context.

- The method is simple and effective, making it more suitable for practical use cases.

- The experiments are well-designed, and the ablations clearly illustrate the impact of various aspects of Remix-DiT.

- The paper is well-written and generally easy to understand.

**Weaknesses:**

- While the authors show the benefits of Remix-DiT on finetuning a pretrained DiT model, it would be interesting to see its effect when training all components from scratch. If the compute budget allows, I suggest that the authors also add this experiment for better insights into what happens if one uses the remixing scheme from the beginning of training (perhaps after a small warmup)

- The performance gain seems to diminish as the size of the base model increases. Hence, a more detailed discussion on this issue is needed for the final version. For example, the performance gain is almost 30% for DiT-S, while it drops to only 15% for DiT-L.


**Minor comments:**

Please fix the following issues in terms of writing in your draft:
- L114 "refer to" -> "refers to"
- L144 -> citation is missing
- L215 -> I assume 100M steps should be 100K steps
- L290 -> it seems that it should be written as N experts because K is the number of base models
- L295 -> "can found" should be "can find"

Please also cite GigaGAN [1] as the mixing part of the paper is related to their method of mixing different convolution kernels during training.

[1] Kang M, Zhu JY, Zhang R, Park J, Shechtman E, Paris S, Park T. Scaling up gans for text-to-image synthesis. InProceedings of the IEEE/CVF Conference on Computer Vision and Pattern Recognition 2023 (pp. 10124-10134).

**Questions:**

1. The EDM paper [1] suggests that for denoising, only the middle noise levels are important, while this paper suggests that the noise levels towards 0 are more crucial. Do you have an intuition on the difference between these two conclusions?

2. Is the performance of Remix-DiT more sensitive to the number of sampling steps compared to a normal DiT? In other words, how do the experts perform when using a deterministic sampler with low NFEs (<50)?

3. Can you also visualize some examples generated by DiT and Remix-DiT? While the metrics are valuable, a qualitative evaluation is interesting as well.

[1] Karras T, Aittala M, Aila T, Laine S. Elucidating the design space of diffusion-based generative models. Advances in neural information processing systems. 2022 Dec 6;35:26565-77.

**Limitations:**

The authors have mentioned this in the paper.

---

> ### Author Rebuttal · Authors · 2024-08-07
>
> > **Q1: While the authors show the benefits of Remix-DiT on finetuning a pretrained DiT model, it would be interesting to see its effect when training all components from scratch. If the compute budget allows, I suggest that the authors also add this experiment for better insights into what happens if one uses the remixing scheme from the beginning of training (perhaps after a small warmup)**
>
> Thanks for the invaluable suggestion. We use pre-trained models to avoid training basis models independently since they share some common ability across timesteps. We agree with the reviewer that it is valuable to explore scratch training with our method. Due to the time limits of this short rebuttal period, we can only provide 400K results for scratch training. It can observed that our method still achieves competitive FID in the setting of scratch training.
> | Model        | FID @ 400K |
> |--------------|------------|
> | DiT-S        | 43.88      |
> | RemixDiT-S   | 36.68      |
>
> > **Q2: The performance gain seems to diminish as the size of the base model increases. Hence, a more detailed discussion on this issue is needed for the final version. For example, the performance gain is almost 30\% for DiT-S, while it drops to only 15% for DiT-L.**
>
> Thank you for the insightful comment! The key motivation behind RemixDiT is that the capacity of a single model is insufficient for multi-step denoising tasks. Therefore, with a smaller model like DiT-S, our method can offer more significant benefits compared to relatively larger models like DiT-L. Additionally, training large models can be more challenging. We will include a more detailed discussion about this phenomenon in the revised version, following the reviewer's advice.
>
> > **Q3: Please fix the following issues in terms of writing in your draft.**
>
> We will revise the draft accordingly.
>
> > **Q4: Please also cite GigaGAN [1] as the mixing part of the paper is related to their method of mixing different convolution kernels during training.**
>
> It's indeed a highly related paper from the perspective of techniques. We will cite it properly.
>
> > **Q5: The EDM paper [1] suggests that for denoising, only the middle noise levels are important, while this paper suggests that the noise levels towards 0 are more crucial. Do you have an intuition on the difference between these two conclusions?**
>
> A large coefficient indicates that this slot is more unique and challenging, necessitating greater capacity to handle it. The segmented patterns in Figure 4(a) demonstrate the importance of both early and intermediate steps. Our method allocates more model capacity to the early stages (0-50) and the intermediate stages (150-500). So, the results is, to some extend, aligned with the observation in the EDM paper.
>
> > **Q6: Is the performance of Remix-DiT more sensitive to the number of sampling steps compared to a normal DiT? In other words, how do the experts perform when using a deterministic sampler with low NFEs (<50)?**
>
> In Table 2 of the submission, we present results for 100-step sampling, where our method continues to show positive results compared to the baseline DiT. In response to your question, we further reduced the number of steps to 25. The following table demonstrates that even with this lower number of NFEs, our method, RemixDiT-S, still outperforms the baseline DiT-S.
> | Model        | FID-10K (Steps=25, cfg=1.5) |
> |--------------|-----------------------------|
> | DiT-S        | 49.82                       |
> | RemixDiT-S   | 44.75                       |
>
> > **Q7: Can you also visualize some examples generated by DiT and Remix-DiT? While the metrics are valuable, a qualitative evaluation is interesting as well.**
>
> Thanks for the suggestion. We supplement visualization results within the attached PDF, which compares the proposed RemixDiT-B to a DiT-B. It can be observed that the shape of objects can be improved using the proposed methods. This is as expected since we allocate model capacity to the early and intermediate stages of denoising, which mainly contributes to the image contents rather than details. We will incorporate visualization results into the experiment sections.

---

> ### Comment · Reviewer_7vev · 2024-08-07
> **Response to the rebuttal**
>
> I would like to thank the authors for taking the time to answer my questions in detail. Since I believe that the role of time step in diffusion networks has been less explored and my concerns have been resolved by the rebuttal, I would like to increase my score from 6 to 7.

---

> > ### Author Response · Authors · 2024-08-07
> >
> > Thank you so much for the positive feedback! We will polish our draft with the above experiments.

---

### Official Review · Reviewer_cagX · 2024-07-10

**Soundness:** 4
**Presentation:** 3
**Contribution:** 4
**Rating:** 7
**Confidence:** 4

**Summary:**

The paper proposes Remix-DiT, a model architecture designed to enhance the capacity of a standard DiT model without significantly increasing inference costs. This is accomplished by training mixing coefficients to adaptively fuse multiple DiT models and developing specialized experts for multi-expert denosing. A key advantage highlighted in this paper is that Remix-DiT achieves better generation quality while maintaining inference speed comparable to that of a standard DiT. Experimental results on ImageNet-256 demonstrate favorable outcomes compared to baseline methods.

**Strengths:**

1.	The visualization results in Figure 4 are very interesting. It seems that the model has a certain preference in allocating the capacity of basis models, with clear segmentation across the timesteps. Additionally, a high coefficient is observed at early timesteps, such as 0-150. Does this imply that those steps are more challenging for the diffusion model to learn?
2.	The idea of mixing multiple basis models is clear and easy to implement. It does not requires the expensive training of independent experts for different steps.

**Weaknesses:**

1.	Using multiple base models may introduce more training costs. However, in Table 3, the GPU memory usage only slightly increases from 13G to 16G for DiT-B. Can the authors provide more details about the reason? Will Remix-DiT introduce a substantial backward and forward footprint?
2.	This method utilizes the pre-trained model as the initialization. This might make the mixed experts always the same after mixing since they are working on the same basis model initially. Will this be a problem?
3.	Why does the proposed method outperform naively training independent experts? In this method, the experts are crafted by mixing, which should theoretically be upper bounded by the naïve method mentioned above.

**Questions:**

Please refer to the weaknesses.

**Limitations:**

This paper discusses limitations such as sparse gradients and the training difficulty associated with a large number of experts.

---

> ### Author Rebuttal · Authors · 2024-08-07
>
> > **Q1: The visualization results in Figure 4 are very interesting. It seems that the model has a certain preference in allocating the capacity of basis models, with clear segmentation across the timesteps. Additionally, a high coefficient is observed at early timesteps, such as 0-150. Does this imply that those steps are more challenging for the diffusion model to learn?**
>
> Thank you for the question. Allocating a high score to a certain slot indicates that this step is distinct from others and might be challenging during training. In the diffusion process, the shape of objects emerges quickly, followed by the incorporation of finer details in the later stages. The algorithm adaptively allocates model capacity to these specific steps, ensuring that critical stages receive the necessary resources for accurate denoising.
>
> > **Q2: This method utilizes the pre-trained model as the initialization. This might make the mixed experts always the same after mixing since they are working on the same basis model initially. Will this be a problem?**
>
> Thanks for the comment, this issue indeed exists at the early training steps. Therefore, we introduce the prior coefficients in Line 188 of the paper to force a prior allocation of model capacity. For example, we initialize the coefficients for the first timestep region with [1,0,0,0], this forces the first basis model to learn from early steps. This trick will lead to diverge basis models, which is important for our method.
>
> > **Q3: Using multiple base models may introduce more training costs. However, in Table 3, the GPU memory usage only slightly increases from 13G to 16G for DiT-B. Can the authors provide more details about the reason? Will Remix-DiT introduce a substantial backward and forward footprint?**
>
> Thank you for the comment; this issue indeed exists in the early training steps. To address this, we introduce prior coefficients, as mentioned in Line 188 of the paper, to enforce a prior allocation of model capacity. For example, we initialize the coefficients for the first timestep region with [1,0,0,0], which forces the first basis model to learn from the early steps. This approach helps diverge the basis models, which is crucial for the effectiveness of our method.
>
> > **Q4: Why does the proposed method outperform naively training independent experts? In this method, the experts are crafted by mixing, which should theoretically be upper bounded by the naïve method mentioned above.**
>
> Thanks for the invaluable question. Our method is more parameter efficient than training $N$ independent experts. First, our method can support training 20 experts simultaneously by optimizing 4 basis models. Under the same budget, our method can be fully optimized while the naive expert training is still underfitted. In addition, our method is able to adaptively allocate model capacity to different timesteps, which improves the utilization of network parameters.

---

### Official Review · Reviewer_2dkC · 2024-07-10

**Soundness:** 3
**Presentation:** 3
**Contribution:** 2
**Rating:** 5
**Confidence:** 4

**Summary:**

To improve the generation quality of diffusion transformers, Remix-DiT proposes to enhance output quality at a lower cost and aims to create N diffusion experts for different denoising timesteps without the need for expensive training of N independent models. Remix-DiT achieves this by employing K basis models (where K < N) and using learnable mixing coefficients to adaptively craft expert models. This approach offers two main advantages: although the total model size increases, the model produced by the mixing operation shares the same architecture as a plain model, maintaining efficiency comparable to a standard diffusion transformer. Additionally, the learnable mixing adaptively allocates model capacity across timesteps, effectively improving generation quality. Experiments on the ImageNet dataset show that Remix-DiT achieves promising results compared to standard diffusion transformers and other multiple-expert methods.

**Strengths:**

Novelty: Model mixers for efficient multi-expert diffusion model training is innovative and unique.

Significance: Addressing the challenge of efficient training of multi-expert diffusion transformers is significant in the field of diffusion models.

Methodology: The proposed algorithm is well-formulated and clearly explained.

Results: Experimental results demonstrate promising improvements over existing methods such as DiT.

**Weaknesses:**

1. Lack of Visualization Results: The paper does not include any visualization results. Providing visual examples of generated outputs is crucial for qualitatively evaluating the effectiveness of the proposed method.

2. Insufficient Motivation for Multi-Expert Training: The rationale behind adopting a multi-expert training approach is not fully well-motivated, particularly in the context of quantitative comparisons. A more detailed explanation of why multi-expert training is beneficial and how it compares quantitatively to other methods would strengthen the argument. Clarifying the advantages and potential trade-offs in performance and efficiency would provide a more compelling case for this approach.

3. High Training Cost: The training cost associated with the proposed method is substantial. It would be beneficial to provide a thorough analysis of the computational resources, time, and energy required for training compared to other existing methods. Discussing potential ways to mitigate these costs or offering insights into why the increased training cost is justified by the performance gains would add valuable context for evaluating the practicality of the method.

**Questions:**

1. Performance Comparison Between Multi-Expert and Single Larger Models: Is it possible for the multi-expert small models to outperform a single, larger model? To fully validate the potential of the multi-expert approach, it is crucial to provide a thorough performance comparison. This should include quantitative metrics and benchmarks that demonstrate the advantages, if any, of using multiple experts over a single larger model in terms of both output quality and computational efficiency.

2. Scalability and Efficiency of Increasing the Number of Experts: If the number of experts is increased for the same basis models, how easily can the system be scaled, and does this lead to more efficient training? It would be important to discuss the scalability of the multi-expert framework, including any potential challenges or limitations in transferring the model to a larger number of experts. Additionally, insights into how the efficiency of training might be affected by increasing the number of experts would be valuable.

**Limitations:**

Please refer to the weakness and question part.

---

> ### Author Rebuttal · Authors · 2024-08-07
>
> > **Q1: Lack of Visualization Results: The paper does not include any visualization results. Providing visual examples of generated outputs is crucial for qualitatively evaluating the effectiveness of the proposed method.**
>
> Thanks for the suggestion. We supplement visualization results in the attached PDF, which compares the proposed RemixDiT-B to a DiT-B. It can be observed that the shape of objects can be improved using the proposed methods. This is as expected since RemixDiT allocates more model capacity to the early and intermediate stages of denoising as illustrated in Figure 4 of the paper, which mainly contributes to the global shape rather than detailed patterns. We will incorporate visualization results into the experiment sections and polish the draft following the advice.
>
> > **Q2: Insufficient Motivation for Multi-Expert Training: The rationale behind adopting a multi-expert training approach is not fully well-motivated, particularly in the context of quantitative comparisons. A more detailed explanation of why multi-expert training is beneficial and how it compares quantitatively to other methods would strengthen the argument. Clarifying the advantages and potential trade-offs in performance and efficiency would provide a more compelling case for this approach.**
>
> The key motivation for the multi-expert approach lies in the limitations of a single model's capacity for the 1000-step denoising tasks. Figure 4 (c) provides a quantitative perspective on this issue by illustrating the performance of experts in both their specialized and non-specialized regions. It can be observed that all experts achieve low losses within their designated time slots, but in the "non-professional" timesteps, an expert may yield relatively large losses. This disparity is due to the limited model capacity when faced with the extensive number of timesteps. To address this, we propose utilizing multi-expert strategies to enhance diffusion models without significantly increasing the number of parameters. We will incorporate the reviewer's advice to clarify the motivation in the revised version.
>
> Based on this, the main advantage of RemixDiT lies in its ability to provide a flexible trade-off between the number of experts and the total training costs through the mixing of a few basis parameters.  This allows training $N$ experts for different regions by optimizing $K (K<N)$ basis models.
>
> > **Q3: High Training Cost: The training cost associated with the proposed method is substantial. It would be beneficial to provide a thorough analysis of the computational resources, time, and energy required for training compared to other existing methods. Discussing potential ways to mitigate these costs or offering insights into why the increased training cost is justified by the performance gains would add valuable context for evaluating the practicality of the method.**
>
> The training cost of RemixDiT can be found in Table 3, which measures the training throughput and memory usage. To obtain 20 experts for different regions, our method incurs 5\% ~ 20\% additional costs compared to a standard DiT. Notably, the training cost does not linearly increase with the number of experts or basis models, as the additional computation primarily arises from the lightweight mixing operation, which will not introduce too much forwarding or backwarding efforts during training. We will revise lines 298-303 of the paper to include a more comprehensive discussion on training and inference efficiency.
>
> > **Q4: Performance Comparison Between Multi-Expert and Single Larger Models: Is it possible for the multi-expert small models to outperform a single, larger model? To fully validate the potential of the multi-expert approach, it is crucial to provide a thorough performance comparison. This should include quantitative metrics and benchmarks that demonstrate the advantages, if any, of using multiple experts over a single larger model in terms of both output quality and computational efficiency.**
>
> Based on our results in the submission, it is challenging for RemixDiT-B with four basis models (FID=9.02) to outperform a DiT-L that is four times larger (FID=3.73), with around 1M training steps. This limitation arises because the number of effective parameters in RemixDiT-B remains equivalent to that of a single DiT-B. Currently, our proposed RemixDiT can ensure better performance under a comparable inference cost. We appreciate the suggestion to explore the upper bound of this method and are working on training DiT baselines of different sizes.
>
> > **Q5: Scalability and Efficiency of Increasing the Number of Experts: If the number of experts is increased for the same basis models, how easily can the system be scaled, and does this lead to more efficient training? It would be important to discuss the scalability of the multi-expert framework, including any potential challenges or limitations in transferring the model to a larger number of experts. Additionally, insights into how the efficiency of training might be affected by increasing the number of experts would be valuable.**
>
> We agree that scalability is important for this method. As shown in Table 2, our key observation is that increasing the number of experts is not always beneficial. There are three reasons for this: 1) As illustrated in Figure 4 (a,b), some denoising steps share similar mixing coefficients, which limits the maximal number of effective experts in our method; 2) The total capacity is also bounded by the number of basis models; 3) As discussed in the limitations (Line 310), optimizing a large number of experts can lead to sparse gradients in the mixing coefficients, making optimization more difficult as the number of experts increases.
>
> Therefore, we chose to use 20 experts in our experiments, which we found to be a good balance between performance and optimization difficulty.

---

### Author Rebuttal · Authors · 2024-08-07

We would like to extend our sincere gratitude to all the reviewers for their time, effort, and insightful feedback on our submission. In response to reviewers' questions, we included some visualization results in the attached PDF file to compare the RemixDiT-B to a standard DiT-B, where our method is able to improve the object shape by allocating more model capacity to early and intermediate timesteps.

---

### Decision · Program_Chairs · 2024-09-25

**Decision:**

Accept (poster)

**Comment:**

The paper presents Remix-DiT, a multi-expert denoising method that enhances diffusion transformers by mixing basis models with learnable coefficients, achieving high-quality outputs without extensive training costs. Reviewers appreciated the novel approach and its potential to efficiently improve generation quality across different timesteps.

While reviewers praised the paper’s practicality and clear presentation, they recommended additional evaluations on diverse datasets and comparisons with state-of-the-art methods like DTR and Switch-DiT. Concerns about high training costs and scalability were noted, along with suggestions to include more visualizations to better showcase the improvements. Addressing these points in the camera-ready would further solidify the paper’s contributions and impact in the field of diffusion models.